# Expression and Characterisation of the First Snail-Derived UDP-Gal: Glycoprotein-N-acetylgalactosamine β-1,3-Galactosyltransferase (T-Synthase) from *Biomphalaria glabrata*

**DOI:** 10.3390/molecules28020552

**Published:** 2023-01-05

**Authors:** Marilica Zemkollari, Markus Blaukopf, Reingard Grabherr, Erika Staudacher

**Affiliations:** 1Department of Chemistry, University of Natural Resources and Life Sciences, 1190 Vienna, Austria; 2Department of Biotechnology, University of Natural Resources and Life Sciences, 1190 Vienna, Austria

**Keywords:** O-glycosylation, T-synthase, *Biomphalaria glabrata*, β-1,3-galactosyltransferase, mollusc

## Abstract

UDP-Gal: glycoprotein-N-acetylgalactosamine β-1,3-galactosyltransferase (T-synthase, EC 2.4.1.122) catalyses the transfer of the monosaccharide galactose from UDP-Gal to GalNAc-Ser/Thr, synthesizing the core 1 mucin type O-glycan. Such glycans play important biological roles in a number of recognition processes. The crucial role of these glycans is acknowledged for mammals, but a lot remains unknown regarding invertebrate and especially mollusc O-glycosylation. Although core O-glycans have been found in snails, no core 1 β-1,3-galactosyltransferase has been described so far. Here, the sequence of the enzyme was identified by a BlastP search of the NCBI *Biomphalaria glabrata* database using the human T-synthase sequence (NP_064541.1) as a template. The obtained gene codes for a 388 amino acids long transmembrane protein with two putative N-glycosylation sites. The coding sequence was synthesised and expressed in Sf9 cells. The expression product of the putative enzyme displayed core 1 β-1,3-galactosyltransferase activity using pNP-α-GalNAc as the substrate. The enzyme showed some sequence homology (49.40% with *Homo sapiens*, 53.69% with *Drosophila melanogaster* and 49.14% with *Caenorhabditis elegans*) and similar biochemical parameters with previously characterized T-synthases from other phyla. In this study we present the identification, expression and characterisation of the UDP-Gal: glycoprotein-N-acetylgalactosamine β-1,3-galactosyltransferase from the fresh-water snail *Biomphalaria glabrata*, which is the first cloned T-synthase from mollusc origin.

## 1. Introduction

O-glycans play an important role in influencing the tertiary structure of proteins, modulating their folding, stability and enzymatic activity. They are involved in protein–protein interaction, recognition processes, take an active part in trafficking of proteins to their target locations and play a role in human health and disease [1,2,3,4]. The biosynthesis of O-glycans is a complex posttranslational event where a number of specific glycosyltransferases are involved. It is started by linkage of one sugar to a specific protein, followed by the sequential addition of other monosaccharides. In contrast to N-glycosylation, this first step of biosynthesis is not restricted to a single organelle but may occur, depending on the linkage, in the ER, Golgi, cytosol or nucleus [3].

Mucin-type O-glycosylation is the most common and best investigated type of O-glycosylation. It starts with the transfer of a GalNAc-residue to a serine or threonine of the polypeptide chain. Depending on the organism, tissue and developmental stage, the protein bound GalNAc residue is further elongated by a GalNAc, a GlcNAc and/or a galactose residue, resulting in eight tissue and function specific subtypes (core structures 1–8). A subsequent modification by the addition of more of these or other monosaccharides (fucose or sialic acid) may occur, forming thousands of different O-glycan species (for a review see [5]).

In core 1 structures the protein bound GalNAc residue is elongated by a glycoprotein-N-acetylgalactosamine β-1,3-galactosyltransferase, also known as T-synthase, forming the disaccharide Gal-β1,3-GalNAc, the T-antigen [1]. A decrease in activity of this enzyme results in a shorter version of the core 1 O-glycan, that lacks the galactose residue, known as Tn-antigen. The Tn-antigen is recognized by Tn antibodies that naturally occur in adult serum. The Tn-antigen plays a role in the development of several human diseases [6]. It is involved in autoimmune disorders (Tn syndrome) and associated with poor prognosis in tumours [7,8]. Mice lacking T-synthase die at embryonic day 14 due to defects in angiogenesis and embryonal haemorrhage [9]. In mammals only one active copy of the *β*-1,3-galactosyltransferase has been found, while in *D. melanogaster* four genes and in *C. elegans* one gene coding for active β-1,3 galactosyltransferases were detected [10,11]. In vertebrates the activity of the enzyme depends on the “**co**re 1 β-Gal-T **s**pecific **m**olecular **c**haperone” (Cosmc) located in the endoplasmic reticulum which binds to the enzyme and thus limits aggregation and degradation during folding [12,13]. The β-1,3-galactosyltransferases in *Drosophila* or *C. elegans* do not need any chaperon for functional activity [10,11].

Molluscs are a large and evolutionary very successful phylum of the animal kingdom. They exist for more than 500 million years with a worldwide occurrence in freshwater, marine and terrestrial habitats and thus, are important members of several ecosystems in terms of “waste disposal” and cleaning but are mainly noticed as a pest in agriculture or as part of the life cycle of parasites [14]. A broad knowledge of their glycosylation abilities may give insights in their extremely successful adaption abilities and surviving strategies under different environmental conditions. For example, mucin-type O-glycosylation has been associated with host–pathogen recognition and in this context the fresh-water snail, *Biomphalaria glabrata* was chosen as an interesting animal model, because it serves as an intermediate host of the pathogen *Schistosoma mansoni* [15].

Previously we could identify and characterise the very first glycosyltransferase from mollusc origin: the initiating enzyme of mucin-type O-glycosylation, the UDP-N-acetyl-α-D-galactosamine: polypeptide N-acetylgalactosaminyltransferase from *Biomphalaria glabrata* [16,17,18]. Here, we present the identification, molecular cloning, expression and characterisation of the subsequent core 1 forming enzyme: the β-1,3-galactosyltransferase (T-synthase) originating from the same snail. It is the first confirmed T-synthase from mollusc origin. A more detailed knowledge of such an important biosynthetic pathway as O-glycosylation will help understand a number of recognition processes in molluscs. So far, nothing is known about the exact function of the enzyme in snails.

## 2. Results

### 2.1. Identification, Cloning and Expression of T-Synthase

The sequence of the *B.glabrata* T-synthase was obtained by a blastP search using the human T-synthase sequence (NP_064541.1) as a template. We identified a 1167 bps long coding sequence (MW720711). The enzyme showed sequence homology with previously characterized T-syntheses from other species (49.40% with *Homo sapiens*, 53.69% *with Drosophila melanogaster* and 49.14% with *Caenorhabditis elegans*) (Figure 1). The presence of conserved motives such as DDD and EDV, further indicated a functional catalytic region. The entire coding sequence was cloned into the pACEBac1 vector and expressed in insect cells (Sf9) cells. The selected gene encoded a 388 amino acids long type II-transmembrane protein (transmembrane region: amino acid 13–30, predicted by TMHMM Server v. 2.0) with two putative N-glycosylation sites (Figure 1). The full-length enzyme containing either a N- or C-terminal His-tag was successfully expressed in Sf9 cells (Appendix A). In order to increase the yield, we tested bacterial and yeast expression systems as an alternative. The ΔTM GalT (truncated version lacking the transmembrane domain) was expressed in BL21 and HMS174 *Escherichia coli* cells and in *Komagatella phaffi* cells but the yield did not improve, so further expression was done exclusively in Sf9 cells. Since there was no difference in yield or binding properties, whether the HisTag was N-ternimally or C-terminally fused to the enzyme, we continued to work with the C-terminal HisTag construct. 

### 2.2. Purification 

None of the tested expression systems yielded secreted β-1,3-galactosyltransferase, even when the transmembrane domain was deleted and the gp64 leader was added for secretion in insect cells or α mating factor was added for secretion in *K. phaffi*. In all cases, the enzyme was only detected in the cell lysate fraction. All attempts to purify the enzyme by a Ni^2+^-NTA column, several types of affinity chromatography (*Lens culinaris* lectin gel, UDP-hexanolamine gel, UDP-agarose, D-Gal-agarose, Ultralink gel coupled with desialylated bovine submaxillary mucin) or by ion exchange chromatography failed. The enzyme either did not bind at all or only to a very small extent. Finally, we succeeded in purifying the recombinant β-1,3-galactosyltransferase by magnetic beads coupled with an anti His Tag antibody. Again, not all of the enzyme could be captured out of the cell lysate, and due to the enzyme’s sensitiviy to the low pH of the elution buffer, it was kept on the beads for the acitivity assay. Yet, the purified T-synthase showed high activity and by repeating the procedure a couple of times we collected an appropriate amount of enzyme.

### 2.3. Specificity Determination of the T-Synthase

The Sf9 cell lysate expressing T-synthase as well as a control Sf9 cell lysate were tested for activity by HPLC (Appendix A). The cell lysate of the cells expressing T-synthase showed high activity levels, while the control cell lysate did not show any activity. Therefore, the β-1,3-galactosyltransferase activity was generated by the expressed enzyme from *B. glabrata* and not by an enzyme of the insect cell expression system. 

NMR-analysis of commercially available pNP-α-GalNAc and pNP-α-GalNAc-β-1,3-Gal was compared with the glycan formed in our activity assay, in order to confirm the formed linkage. Due to very limited substance amounts, the main body of analysis was carried out by ^1^H NMR. Comparison of the anomeric region with pNP-α-GalNAc reference material showed that within the sample only one additional substance was present, which was identical to the commercially available product pNP-α-GalNAc-β- 1,3-Gal. Further analysis via COSY and HSQC experiments showed position 3 of GalNAc to be the site of glycosylation. The large coupling constant of the anomeric galactose signal (J1,2 = 7.7 Hz) further showed the linkage to be β-(1→3) which was also in agreement with previously reported analytical data [19] (Appendix A). Therefore, the identity of the product pNP-α-GalNAc-β-1,3-Gal was clearly confirmed. To investigate the substrate specificity of the expressed enzyme, other pNP-labelled monosaccharides (pNP- α-Gal, pNP- α-Glc, pNP-β-GalNAc, pNP-β-Gal, pNP- β-Glc and pNP- β-GlcNAc.) were tested. However, none of these was suitable as substrate for the T-synthase from *B. glabrata*. Thus, exclusively the monosaccharide GalNAc in α-position was found to be a functional substrate. An α-GalNAc structural element is a necessary but not sufficient acceptor. For example, benzyl-α-GalNAc was not galactosylated. Furthermore, the peptides α-GalNAc-Muc2, α-GalNAc-Muc5Ac and α-GalNAc-CHT3, which were beforehand monoglycosylated by UDP-N-acetyl-α-D-galactosamine: polypeptide-N-acetylgalactosaminyl-transferase from *Biomphalaria glabrata* were tested according to [17]. 

While α-GalNAc-Muc2 or α-GalNAc-Muc5Ac were not galactosylated, α-GalNAc-CHT3 turned out to be a suitable substrate for the enzyme.

### 2.4. Biochemical Properties of the Enzyme

The activity of the T-synthase was not affected by storage at temperatures ranging from −80 °C to 37 °C for 24 h (Figure 2a). Additionally, a concentration of up to 15% of methanol, acetonitrile or glycerol did not drastically affect the activity. 37 °C was the optimal incubation temperature for the activity reaction assay for up to 2 h (Figure 2b). Moreover, β-1,3- galactosyltransferase showed a pH optimum at 6.5 as tested by MES as salt buffer (Figure 3a).

To determine the importance for divalent cations, the standard activity assay was carried out without any addition of cations and in the presence of 20 mM EDTA, Mn^2+^, Mg^2+^, Ca^2+^, Co^2+^, Cu^2+^, Ni^2+^, or Ba^2+^. The enzymatic reaction was dependent on divalent cations, as in the presence of EDTA no transfer was observed, while in the presence of Mn^2+^ maximum transfer rates were observed (Figure 3b). Co^2+^ slightly increased the activity, Mg^2+^ and Ca^2+^ had no effect, while Ba^2+^, Cu^2+^ and Ni^2+^ had negative effects on T-synthase activity. Maximal rates of transfer were obtained with a Mn^2+^ concentration of 25 mM (Figure 3c). The addition of 5 nmols of UMP, UDP or UTP to the standard assay had an inhibitory effect on the activity of the enzyme of about 50%. Addition of 5 nmols GDP decreased the activity by 33%, while an addition of galactose, glucose or N-acetylgalacosamine had no significant effect.

## 3. Discussion 

UDP-Gal: glycoprotein-N-acetylgalactosamine β-1,3-galactosyltransferase (T-synthase), an elongation enzyme in the pathway of forming mucin type core I O-glycans, was identified in the genome sequence of *B. glabrata* by sequence homology and as the first mollusc derived enzyme of its kind recombinantly expressed, purified and characterised in detail. Due to the relative low expression yield in insect cells, purification of the T-synthase became quite challenging. Attempts to purify it via metal chelate affinity chromatography, ion exchange chromatography or different kinds of affinity chromatography failed. We finally succeeded to purify the enzyme by precipitation, using magnetic beads coupled with anti His Tag antibodies.

Alignment studies showed that β-1,3-galactosyltransferase from *B. glabrata* has some similarities with previously characterised homologous enzymes from other species (49.40% with *Homo sapiens*, 53.69% with *Drosophila melanogaster* and 49.14% with *Caenorhabditis elegans*). Moreover, the snail enzyme comprises well described conserved sequence motifs (DDD and EDV in this case) in the middle of the protein, which are known to be involved in binding of the substrate, the sugar donor and cations. Glycosyltransferases containing a DXD motif use UDP-sugars as donors and need a divalent cation for their activity [20].

The *B. glabrata* T-synthase also contains the CCSD sequence close to the C-terminus. This is an identification motif for all T-synthases and other associated proteins along with Cosmc [11]. Cosmc is the chaperon that vertebrate T-synthases (human, rat) require for appropriate folding. In contrast, corresponding enzymes in invertebrates (*D. melanogaster, C. elegans, B. glabrata*) do not need chaperons [10,11,21,22]. At least, so far, no Cosmic chaperon homolog has been found in any of the know invertebrate genomes.

All previously described T-synthases share six conserved cysteines in the catalytic region. In addition, similar to other invertebrate T-synthases, the one from *B. glabrata* has a seventh cysteine residue in the catalytic domain. The mammalian enzymes have also two cysteins in the transmembrane domain, while the invertebrate enzymes lack these residues. *D. melanogaster* T-synthase is an exception as it has, similar to the other invertebrates, seven cysteins in the catalytic domain, but also one in the transmembrane domain.

In general, N-glycosylation does not seem to play a major role in the activity of T-synthases. The sequence of the recombinant snail β-1,3-galactosyltransferase predicts two putative N-glycosylation sites, one of those in the catalytic domain. The sequence of the human core 1 β-1,3-galactosyltransferase predicts zero and that of *C. elegans* predicts four N-glycosylation sites [11,21].

Along with the structural similarities with the previously described T-synthases, *B. glabrata* T-synthase also shows biochemical similarities to them. It has an optimum in activity at pH 6 and the activity is dependent on the presence of divalent cations, preferably manganese. Addition of EDTA completely abolishes the activity. The enzyme is stable when incubated at temperatures ranging from −80 °C to 37 °C overnight and it is active in a wide temperature range from 4 °C to 50 °C. pNP-α-GalNAc and α-GalNAc-CHT3 peptides are galactosylated by the *B. glabrata* T-synthase but benzyl-α-GalNAc and Muc2-α-GalNAc or Muc5Ac-α-GalNAc peptides are not. This means that even though the enzyme is able to recognise α bound GalNAc residues, the surroundings of the offered GalNAc-substrate has a huge influence. This observation supports the study of Perrine on the acceptor specificity of human T-synthase, which shows an influence of different amino acids near the glycosylation site. This study shows that not only the first, but also the second enzyme of the O-glycosylation pathway (T-synthase) is still influenced by the structure of the peptide. For example, basic amino acids close to the glycosylation site prevent a transfer of galactose [23]. It is possible that the two Gly residues in + 3 and −3 position and the two Pro residues in +2 and −2 position relative to the acceptor glycosylation side of the snail enzyme are responsible for α-GalNAc-CHT3-peptide being a suitable acceptor, in contrast to α-GalNAc-Muc2-peptide and α-GalNAc-Muc5Ac-peptide, which are not.

## 4. Materials and Methods

### 4.1. Materials

Restriction enzymes, T4 ligase, Q5 DNA polymerase and OneTaq DNA Polymerase were all purchased from New England Biolabs (Frankfurt, Germany). Plasmid purification and Gel clean-up kits were purchased from Macherey-Nagel (Düren, Germany). The g-Block gene fragment was synthetized by Integrated DNA Technologies (Leuven, Belgium). All primers were commercially synthetized by Integrated DNA Technologies or Sigma Aldrich (Vienna, Austria).

Antibodies: goat 6xHis epitope antibody (Thermo Fischer Scientific, Vienna, Austria), dilution 1:3000, alkaline phosphatase conjugated anti-goat IgG (Thermo Fischer Scientific, Vienna, Austria) dilution 1:4000.

Chemically competent *Escherichia coli* (High Efficiency) cells (New England Biolabs, Frankfurt, Germany) were spread on Lysogeny broth (LB) agar plates containing 15 μg/mL Gentamicin (NEB5α). Positive constructs were electrocorporated into DH10 Multibac YFP E. coli cells (Geneva Biotech, Geneva Switzerland) and spread on LB Agar plates containing 15 μg/mL Gentamicin, 100 μg/mL Kanamycin, 100 μg/mL Tetracycline, 100 μg/mL X-Gal and 100 μg/mL IPTG and incubated overnight at 37 °C. The electroporation was performed at 2500 V, 25 µF and 200 Ω (MicroPulser from BIORAD).

*Spodoptera frugiperda* cells (Sf9, ATCC CRL-1711, Manassas Virginia) were cultivated in IPL41 medium (HyClone Cytiva—Vienna, Austria) containing yeast extract, a lipid mixture supplemented with 10% fetal calf serum at 27 °C.

Graphs were prepared with Graph Pad Prism 8.0.

### 4.2. Identification of the Gene Sequence 

The protein sequence of the T-synthase from *B. glabrata* was obtained by performing a blastP search, using the sequence of the human homologue, T-synthase, protein sequence (NP_064541.1, downloaded on 01.09.2020) as the template. The putative coding sequence obtained from NCBI was codon optimized for insect cells (genescrip.com) and was synthetized by Integrated DNA Technologies (Leuven, Belgium). The nucleotide sequence reported in this paper has been deposited in GenBank database with the accession number MW720711. Alignment studies were done using Clustal Omega program. 

### 4.3. Cloning and Expression of the Synthetic Sequence

The optimized gene sequence was cloned into the pACEBac1 vector. Four distinct constructs were created: C-terminus His Tag construct, gp64 Leader sequence construct with a C-terminus His Tag, N-Terminus His Tag construct and a soluble construct that has the 6xHisTag fused to the 47th amino acid residue. The following primers were used to amplify each insert fragment, respectively: 5′ TAT TAT GAA TTC ATG GCC CCA ATC AGT CAC and 5′ GAT GAT TCT AGA TTA ATG GTG GTG ATG ATG TCC AG; 5′ GAT GAT GAA TTC ATG GTC AGT GCC ATT and 5′GAT GAT TCT AGA TTA GTG GTG ATG GAG AT; 5′ GAT GAT GAA TTC ATG CAT CAT CAC CAC CAC CAT GGA ATC CGG GCC CCA ATC AGT CAC and 5′ TAT TAT TCT AGA TTA GGA GAT GTG TTT AGA ATC AGT TTG; 5′ GAT GAT GAA TTC ATG CAT CAT CAC CAC CAC CAT GGA ATC CGG GAC TCC CCC CAT TCC and 5′ TAT TAT TCT AGA TTA GGA GAT GTG TTT AGA ATC AGT TTG.

The pACEBac1 vector and each of the insert DNA fragment were digested with EcoRI and XbaI. Restriction digestion was performed at 37 °C for 1 h and then the enzymes were inactivated for 15 min at 65 °C. Ligation with T4 DNA ligase was performed overnight at 16 °C. To confirm the sequence and the correct insertion, all the constructs were sent for sequencing (Microsynth,). 2 μg of each construct were electrocorporated into DH10 MultiBac YFP cells. The incorporation of the construct into the baculovirus DNA by site specific transposition, interrupts a LacZ gene, so in a plate containing X-Gal and IPTG the positive colonies will appear white. The recombinant baculovirus DNA was isolated from the white positive colonies and 5 μg of it was used to transfect Sf9 cells [24]. For the production of the seed stock 0.9 × 10^6^ cells in 2 mL of medium were used. After they were attached to the surface of the well, 5 µg bacmid in 200 μL solution containing 9 μL FuGene and medium, was carefully added to the cells.

Transfected cells were incubated at 27 °C for 5 days. After 5 days the seed stock was harvested and used to produce the intermediate stock. 8 × 10^6^ cells in a 12 mL medium were used for the production of intermediate sock. 360 µLof FBS and 100 μL of the seed stock were added carefully.

The cells were incubated for 4 days at 27 °C. After 4 days the intermediate stock was collected and used for the production of the working stock. 2 × 10^6^ cells/mL in a total volume of 50 mL medium were used. 2 mL of FBS and 200 µL intermediate stock were added. The cells were incubated at 27 °C, shaking, for 4 days. The working stock then was used for the production of the recombinant protein. 1.5 × 10^6^ cells/mL in a volume of 50 mL were infected with baculoviral DNA (MOI 10). The protein was collected 3 days post infection.

Expression in *K. phaffi* was done using a Golden Gate cloning system with $P_{AOX1}$ or $P_{G1}$ as promoters and ScCYC1 as terminator [25].

Expression in *E. coli* was done using a pet-30a plasmid in BL21(DE3) (Thermo Fisher) and in HMS174(DE3) (Sigma-Aldrich). The expression of the protein was induced by Isopropyl-β-D-1-thiogalactopyranosid (25 mg/mL) (Sigma-Aldrich).

### 4.4. Protein Detection and Purification

The recombinant protein was isolated by lysing the insect cells with 400 mM MES buffer pH 6.5, using an Ultraturrax. The homogenate was centrifuged for 3 min at 16,000 rpm at 4 °C. The supernatant was removed to new pre-cooled Eppendorf tubes and saved at 4 °C for further analysis.

For SDS-PAGE 400 µL of SF9 supernatant/cells lysates were firstly methanol precipitated by adding 4× volume ice-cold methanol and incubating them for 30 min at −80 °C. After centrifugation at 16,000× *g* for 30 min at 4 °C, the supernatant was discarded, the pellets air dried and resuspended in 8 μL of water. An equal volume of 2× sample buffer (31 mg DTT, 200 mg SDS, 5 mL 0.5 M Tris/HCl pH 6.8, 2.8 mL glycerol (87%), 2.7 mL H_2_O, a few drops of bromphenolblue) was added to the samples and they were incubated at 99 °C for 5 min before loading them on a 12.5% acrylamide gel. The sample was analysed by SDS-PAGE and Western Blot using mouse 4xHis epitope antibody, (Qiagen, dilution 1:2000) followed by alkaline phosphatase conjugated anti-mouse IgG (Sigma-Aldrich, Vienna, Austria; dilution 1:4000). Bovine serum albumin was used as a negative control for 4xHis epitope monoclonal antibody.

For activity assays the purification was performed using magnetic beads coupled with an anti His Tag antibody (Abbkine, A02050MGB), according to the manufacturer’s instructions.

### 4.5. Activity Assay of the Recombinant T-Synthase

The activity of the recombinant β-1,3-galactosyltransferase was measured in a total volume of 20 μL reaction mixture containing: 50 mM MES pH 6.5, 20 mM $MnCl_{2}$, 200 μM UDP-Gal (Sigma-Aldrich, Vienna, Austria), 1 mM pNP-α-GalNAc, 2 mM ATP and 5 μL enzyme at 37 °C for 2 h. The reaction was terminated by freezing the sample at −20 °C for 5 min and analysed by HPLC on a reversed phase C18 column 4 × 250 mm, 5 μL (Thermo Scientific, Vienna, Austria) in 0,1 M ammonium acetate pH 6.0 (solvent A) applying a linear gradient with solvent B (acetonitrile in water 50:50) from 5 to 50% in 30 min at a flow rate of 1 mL/min. For the analysis of the biochemical parameters the standard assay conditions were modified as follows. For the determination of cation requirements, the assay was carried out without any cations or in presence of 10 mM EDTA, $Mn^{2+}$,$\;Mg^{2+}$, $Ca^{2+}$, $Ba^{2+}$, $Ca^{2+}$, $Cu^{2+}$ and $Ni^{2+}$. To determine the optimum $Mn^{2+}$, the concentration of $MnCl_{2}$ was varied from 0–30 mM. The determination of enzyme stability and pH optimum was done according to Peyer et al. [26]. Substrate specificity was tested with the artificial substrates pNP- α-Gal, pNP- α-Glc, pNP-β-GalNAc, pNP-β-Gal, pNP- β-Glc and pNP-β-GlcNAc, benzyl-α-GalNAc and the α-GalNAc containing peptides PTTTPITTTTTVTPTPTPTGTQTK (α-GalNAc-Muc2), GTTPSPVPTTSTTSAP (α-GalNAc-Muc5Ac) and APPAHPGPTPGYRPAPG (α-GalNAc-CHT3) under standard assay conditions. Inhibition experiments were carried out by adding 5 nmols of of UMP, UDP, UTP, GDP, Gal, GalNAc or Glc to the standard incubation assay.

### 4.6. Determination of Protein Content

To determine the concentration of the proteins, Micro-BCA protein assay (Pierce, Bonn, Germany) was used with bovine serum albumin as the standard.

### 4.7. Matrix Assisted Laser Desorption Ionisation—Time of Flight (MALDI-TOF) Mass Spectrometry

MALDI-TOF MS analysis was carried out on an Autoflex Speed MALDI-TOF (Bruker Daltonics, Germany) equipped with a 1000 Hz Smartbeam.II laser in positive mode using α-cyano-4-hyroxycinnamic acid as matrix (1% *w*/*v* in 65% *v*/*v* acetonitrile solution). For crystallization 1 μL of an 1:40 dilution of the samples was spotted on the plate, air dried, covered by 1 μL of matrix solution and again air dried. Spectra were processed with the manufacturer’s software (Bruker Flexanalysis 3.3.80).

### 4.8. NMR Analysis

NMR spectra were recorded with a Bruker Avance III 600 instrument (600.22 MHz for ^1^H, 150.93 MHz for ^13^C) using standard Bruker NMR software. ^1^H spectra were recorded in D_2_O at 300 K. Assignments were based on COSY, HSQC, HMBC data.

## 5. Conclusions

In this study, we present the first cloned T-synthase from mollusc origin. UDP-Gal: glycoprotein-N-acetylgalactosamine β-1,3-galactosyltransferase from the fresh-water snail *Biomphalaria glabrata* is identified, expressed, purified and characterised. This enzyme catalyses the second step of core 1 O-glycan biosynthesis, forming T-antigen structures. Consistent with the T-synthases from other invertebrates, the mollusc enzyme is not dependent on a chaperone such as Cosmic. Otherwise, the enzyme shows similar biological parameters to all known T-synthases. This study is further proof of the precision and high comparability of glycosylation pathways across all species.

## Figures and Tables

**Figure 1 molecules-28-00552-f001:**
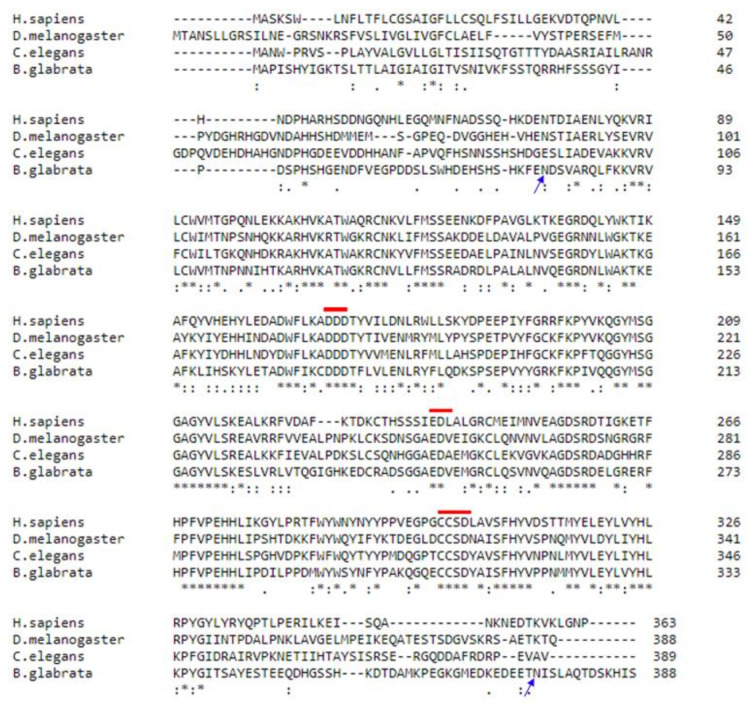
Comparison of the T-synthases from different species. The conserved motives are indicated by red bars and putative N-glycosylation sites are indicated by blue arrows. (***** fully conserved residues, **:** residues with strongly similar properties, **.** residues with weakly similar properties, - indicates gap).

**Figure 2 molecules-28-00552-f002:**
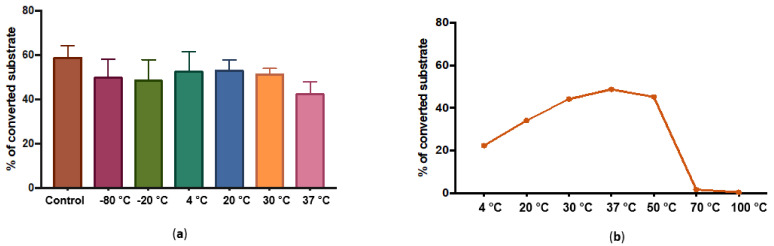
Effect of temperature on enzymatic activity (**a**) Effect of incubation at different temperatures; (**b**) Effect of overnight storage in different temperature on activity.

**Figure 3 molecules-28-00552-f003:**
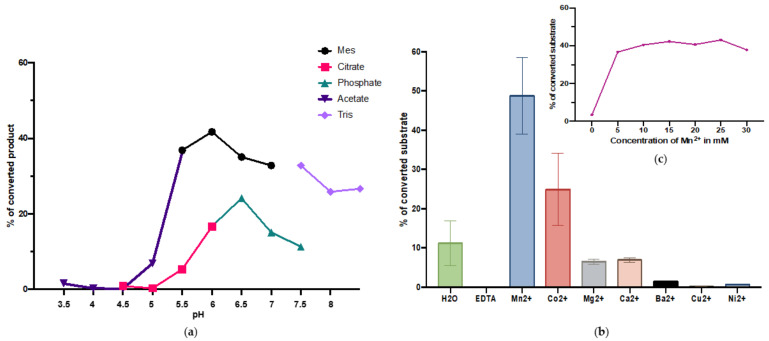
Biochemical properties of the recombinant T-synthase (**a**) Effects of different buffers and pH on the activity; (**b**) Influence of divalent cations on the enzyme activity; (**c**) Manganese concentration curve.

## Data Availability

Not applicable.

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
