# Peer review of "Expression and Characterisation of the First Snail-Derived UDP-Gal: Glycoprotein-N-acetylgalactosamine β-1,3-Galactosyltransferase (T-Synthase) from Biomphalaria glabrata"

_molecules, 2023, doi:10.3390/molecules28020552_

Round 1

Reviewer 1 Report

This manuscript describes the identification, expression and characterisation of the UDP-Gal: glycoprotein-N-acetylgalactosamine β-1,3-galactosyltransferase from the fresh water snail Biomphalaria glabrata, which is the first cloned T-synthase from mollusc origin.

1. English writing should be standardized and more consistent. Many errors could be found. For example, authors always use commas instead of decimal points, such as 49,40%, 53,69% and 49,14% in the abstract part.

2. In lines 133-134, “fully con-133 served residue, : residues with strongly similar properties, . residues” showed two punctuation marks in one position twice.

3. There should be a space after the colon in "UDP-Gal:glycoprotein-N-acetylgalactosamine".

4. Regarding  "For example;" In line 145, The semicolon is also wrong.

5. There should be a space between the number and unit, for example, "-80°C to 37°C" in line 217 and "4°C to 50°C".

6. Microlitre should be "mL", not "ml".

7.  In line 388, two figure captions appeared in the same position: "Figure S3: Figure S2: 1H NMR of the incubation..."

8. There are also many errors in the reference part, for example, "Angew. Chem. Int. Ed. Engl.", "PNAS" and "Acta-Gen. Subj" are not standard abbreviations.

Author Response

The authors thank the reviewers for their helpful comments and suggestions. Please find enclosed in detail our changes regarding each comment.

Reviewer 1

  1. English writing should be standardized and more consistent. Many errors could be found. For example, authors always use commas instead of decimal points, such as 49,40%, 53,69% and 49,14% in the abstract part.
  2. In lines 133-134, “fully con-133 served residue, : residues with strongly similar properties, . residues” showed two punctuation marks in one position twice.
  3. There should be a space after the colon in "UDP-Gal:glycoprotein-N-acetylgalactosamine".
  4. Regarding  "For example;" In line 145, The semicolon is also wrong.
  5. There should be a space between the number and unit, for example, "-80°C to 37°C" in line 217 and "4°C to 50°C".
  6. Microlitre should be "mL", not "ml".
  7. In line 388, two figure captions appeared in the same position: "Figure S3: Figure S2: 1H NMR of the incubation..."
  8. There are also many errors in the reference part, for example, "Angew. Chem. Int. Ed. Engl.", "PNAS" and "Acta-Gen. Subj" are not standard abbreviations.

All mentioned mistakes – and some more - regarding standardization and journal abbreviations  were corrected.

Reviewer 2 Report

Comments to authors

The authors have contributed the first T-synthase from snail, a mollusk origin. T-synthase is important for the synthesis of Tn-antigens , which play a crucial role in human diseases. Although this T-synthase is from snail, to build up the entire protein database (for homology) is also important for comparison and developing further approach for fighting any diseases in related area. The authors did systematically characterize this protein from enzymatic activity to biochemical properties…etc. HPLC and NMR analysis undoubtedly indicated that T-synthase did produce the desired product. From my point of view, this article is benefit for biochemists, who are working on the related areas. Therefore, it is worthy of publication in Molecules.

However, minor revision is also necessary before it could be published with the permission of the chief editor.

1)      Please emphasize how important the T-synthase form mollusc origin “A LITTLE MORE” in the introduction. It may attract more attention from the readers. Is this T-synthase related to any diseases of snail? If yes, please indicate what. If not, it’s OK.

2)      Please compared the snail T-synthase and human T-synthase, especially the enzymatic activity. This may provide a valuable data for medicinal chemists to develop enzyme inhibitors or anticancer agents.

Author Response

The authors thank the reviewers for their helpful comments and suggestions. Please find enclosed in detail our changes regarding each comment.

Reviewer 2

  • Please emphasize how important the T-synthase form mollusc origin “A LITTLE MORE” in the introduction. It may attract more attention from the readers. Is this T-synthase related to any diseases of snail? If yes, please indicate what. If not, it’s OK.

An additional sentence was added to the introduction to emphasize the enzyme.

So far, there is no information regarding O-glycosylation in context with diseases of snails. 

2.)  Please compared the snail T-synthase and human T-synthase, especially the enzymatic activity. This may provide a valuable data for medicinal chemists to develop enzyme inhibitors or anticancer agents.

The last paragraph of the results was modified and extended. However, also for the human enzyme not much is known about substrate specificity in detail. Therefore, any specificity comparison is highly speculative.

This observation supports the study of Perrine on the acceptor specificity of human T-synthase, which shows an influence of different amino acids near the glycosylation site. This study shows that not only the first, but also the second enzyme of the O-glycosylation pathway (T-synthase) is still influenced by the structure of the peptide. For example, basic amino acids close to the glycosylation site prevent a transfer of galactose [23].  It is possible that the two Gly residues in + 3 and -3 position and the two Pro residues in +2 and -2 position relative to the acceptor glycosylation side of the snail enzyme are responsible for α-GalNAc-CHT3-peptide being a suitable acceptor, in contrast to α-GalNAc-Muc2-peptide and α-GalNAc-Muc5Ac-peptide, which are not. 
